# Disentangling Test-Time and Parameter Scaling for Cost-Efficient Accuracy Improvements in Agentic Evaluation

## Abstract

Large language models (LLMs) offer two primary levers for improving accuracy in agentic systems: *test-time scaling* (e.g., Chain-of-Thought reasoning) and *parameter scaling* (upgrading to larger models). Despite widespread adoption, the field lacks principled evaluation of the accuracy-cost-latency trade-offs under controlled conditions. We present a comprehensive evaluation framework and conduct experiments on GSM8K (1,319 items) and PopQA (2,000-item subset) to establish these trade-offs. Our key findings reveal that: (i) on mathematical reasoning tasks, Chain-of-Thought is highly effective for smaller models but becomes redundant when internal reasoning capabilities are available; (ii) on knowledge-intensive QA, performance is primarily capacity-bound, with Chain-of-Thought often increasing costs without improving accuracy; (iii) for models with advanced reasoning capabilities, external Chain-of-Thought becomes largely redundant and can even harm performance while increasing costs. We formalize Pareto frontiers and cost-per-point metrics that translate into actionable deployment policies for more efficient agentic systems.

## 1 Introduction

Modern agentic LLM systems increasingly handle complex scientific workflows, from hypothesis generation to experimental design and manuscript drafting. As these systems evolve from assistants to autonomous actors, practitioners face a fundamental optimization question: when performance is insufficient, should they apply Chain-of-Thought (CoT) [13] prompting to enhance reasoning, or upgrade to a more capable model?

The former approach increases inference-time computation through explicit step-by-step reasoning but inflates token usage and latency. The latter improves underlying capacity but typically increases per-token costs. While teams routinely make these trade-offs, systematic evidence comparing their cost-effectiveness across different task domains remains limited.

This paper advocates for a domain-aware, budget-conscious approach to model selection. We hypothesize that different task types fail for fundamentally different reasons. Multi-step mathematical problems are often *reasoning-limited*, where errors stem from computational mistakes or inadequate problem decomposition—issues that explicit reasoning chains can address. Conversely, open-domain factual QA tasks are typically *knowledge-limited*, where failures arise from missing or imprecise information that longer reasoning chains cannot remedy [8].

We develop a controlled evaluation framework that systematically separates external Chain-of-Thought prompting from internal reasoning capabilities (via model-native controls) while simultaneously measuring accuracy, monetary cost, and latency. Our experiments span two complementary

domains: GSM8K for mathematical reasoning [3] and PopQA for the retrieval of factual knowledge [9].

**Research Hypotheses.** **H1:** On mathematical tasks, Chain-of-Thought provides substantial benefits for smaller models but offers diminishing returns when internal reasoning is available. **H2:** On knowledge-intensive tasks, parameter scaling dominates Chain-of-Thought in both accuracy gains and cost efficiency. **H3:** For models with internal reasoning capabilities, external Chain-of-Thought becomes largely redundant and can even harm performance while increasing costs.

**Contributions.**

- **Methodological framework:** A controlled comparison isolating external Chain-of-Thought from internal reasoning to identify when they are substitutable versus complementary.
- **Cost-latency analysis:** Comprehensive logging of per-sample costs and latencies alongside accuracy, with metrics for cost-per-percentage-point improvements and Pareto frontier analysis.
- **Deployment guidance:** Evidence-based recommendations for practitioners: prioritize parameter scaling initially, then apply reasoning techniques selectively based on task characteristics.

# 2 Related Work

## 2.1 Test-Time Scaling Techniques

Chain-of-Thought prompting [13] elicits intermediate reasoning steps through few-shot examples or explicit instructions, demonstrating particular effectiveness on arithmetic and logical reasoning tasks [7]. Extensions include Program-of-Thoughts [2], which leverages code execution for mathematical reasoning, and Tree-of-Thoughts [14], which explores multiple reasoning branches. Self-consistency approaches [12] sample multiple reasoning paths and select the most frequent answer.

These techniques consistently improve performance on algorithmic tasks but incur computational overhead through increased token generation, longer inference times, and potential verbosity that can harm exact-match extraction in retrieval-oriented tasks.

## 2.2 Parameter Scaling and Model Capacity

The scaling hypothesis suggests that larger models with more parameters generally achieve better performance across diverse tasks [6, 5]. Recent work has explored the trade-offs between model size and inference efficiency [11], particularly in deployment scenarios with strict latency or cost constraints.

However, systematic comparisons of parameter scaling versus test-time techniques under matched experimental conditions remain limited, particularly when controlling for domain-specific task characteristics.

## 2.3 Internal Reasoning and Thinking Tokens

Recent model architectures incorporate internal reasoning capabilities that allocate additional computation without generating observable tokens [15]. These systems can perform latent reasoning similar to explicit Chain-of-Thought but with different cost and latency profiles [4]. Our work leverages these capabilities where available to isolate the effects of reasoning from token generation overhead.

## 2.4 Cost-Aware LLM Evaluation

While most benchmark evaluations focus primarily on accuracy metrics, recent work has begun incorporating cost and latency considerations for practical deployment scenarios [1, 10]. Our approach extends this line of work by providing systematic cost-per-improvement metrics and Pareto frontier analysis for strategic decision-making.

## 3 Methodology

### 3.1 Experimental Framework

For a dataset $D = \{(x_i, a_i^*)\}_{i=1}^N$ with inputs $x_i$ and ground truth answers $a_i^*$, we define strategy $s \in \{\text{No-CoT}, \text{CoT}\}$, model family $f$, and capacity level $m$. Our evaluation metrics are:

$$A(f, m, s) = \frac{1}{N} \sum_{i=1}^N \mathbf{1}[g(y_i(f, m, s)) = a_i^*] \tag{1}$$

$$C(f, m, s) = \frac{1}{N} \sum_{i=1}^N \text{cost}(y_i) \tag{2}$$

$$L(f, m, s) = \frac{1}{N} \sum_{i=1}^N \text{latency}(y_i) \tag{3}$$

where $g(\cdot)$ is an extraction function, $y_i$ is the model output, and we measure accuracy $A$, cost $C$, and latency $L$. We report improvements relative to baseline configurations and calculate cost-per-percentage-point (CPP) as $\Delta C / \Delta A$.

### 3.2 Task-Specific Prompting

**GSM8K (Mathematical Reasoning).** We employ an 8-shot Chain-of-Thought template that enforces deterministic answer extraction:

```
Question:  Janet's ducks lay 16 eggs per day...
Reason step by step, then give the final numeric answer.
Final line must be:  #### <number>
Answer:  Step 1:  Calculate...  Step 2:  Therefore...
#### 18
```

The No-CoT variant requests only the final numeric answer in the specified format.

**PopQA (Knowledge Retrieval).** The Chain-of-Thought template elicits brief reasoning while ensuring clean answer extraction:

```
Question:  Who wrote "Pride and Prejudice"?
Answer step by step in 1-3 sentences, then provide:
Final Answer:  <entity>
```

The No-CoT variant requests only the direct answer. We evaluate using exact match (EM) and F1 scores with standard normalization.

### 3.3 Internal Reasoning Controls

For models supporting internal reasoning controls (specifically Gemini Flash), we manipulate the thinking budget parameter: setting it to zero disables internal reasoning, while omitting the parameter enables it. This allows us to isolate capacity effects from reasoning effects while maintaining identical prompts and decoding parameters.

### 3.4 Experimental Controls

All experiments use consistent random seeds (42), temperature 0.7, top-p 1.0, and appropriate timeouts. We log input/output tokens, end-to-end latency, and per-sample costs. Self-consistency sampling is disabled due to API reliability constraints, which we discuss in our limitations.

Table 1: Complete GSM8K results across all model configurations.

| Model | Reasoning | CoT | Accuracy | Cost/Sample ($) | Total Cost ($) | Latency/Sample (s) |
|---|---|---|---|---|---|---|
| GPT-4.1-mini | N/A | No | 45.19% | 0.000042 | 0.055 | 0.65 |
| GPT-4.1-mini | N/A | Yes | 95.15% | 0.000763 | 1.006 | 2.29 |
| GPT-4.1 | N/A | No | 57.01% | 0.000208 | 0.275 | 0.75 |
| GPT-4.1 | N/A | Yes | 94.69% | 0.003889 | 5.130 | 2.55 |
| Gemini 2.5 Flash-Lite | N/A | No | 36.67% | 0.000025 | 0.033 | 0.80 |
| Gemini 2.5 Flash-Lite | N/A | Yes | 93.85% | 0.000250 | 0.330 | 1.45 |
| Gemini 2.5 Flash | Disabled | No | 55.19% | 0.000038 | 0.050 | 0.58 |
| Gemini 2.5 Flash | Disabled | Yes | 95.60% | 0.000944 | 1.246 | 1.45 |
| Gemini 2.5 Flash | Enabled | No | **95.36%** | 0.000038 | 0.049 | 2.07 |
| Gemini 2.5 Flash | Enabled | Yes | 95.27% | 0.000889 | 1.172 | 2.99 |
| Gemini 2.5 Pro | Enabled | No | 96.18% | 0.000154 | 0.203 | 7.57 |
| Gemini 2.5 Pro | Enabled | Yes | 96.41% | 0.003867 | 5.100 | 10.88 |

Table 2: Complete PopQA results across all model configurations.

| Model | Reasoning | CoT | Accuracy | Cost/Sample ($) | Total Cost ($) | Latency/Sample (s) |
|---|---|---|---|---|---|---|
| GPT-4.1-mini | N/A | No | 36.05% | 0.000022 | 0.045 | 0.64 |
| GPT-4.1-mini | N/A | Yes | 40.85% | 0.000201 | 0.402 | 1.99 |
| GPT-4.1 | N/A | No | **49.55%** | 0.000110 | 0.220 | 0.74 |
| GPT-4.1 | N/A | Yes | 44.55% | 0.000911 | 1.822 | 2.40 |
| Gemini 2.5 Flash-Lite | N/A | No | 29.50% | 0.000005 | 0.009 | 0.70 |
| Gemini 2.5 Flash-Lite | N/A | Yes | 29.59% | 0.000046 | 0.093 | 1.13 |
| Gemini 2.5 Flash | Disabled | No | 33.50% | 0.000018 | 0.035 | 0.56 |
| Gemini 2.5 Flash | Disabled | Yes | 38.95% | 0.000208 | 0.416 | 0.94 |
| Gemini 2.5 Flash | Enabled | No | **40.13%** | 0.000018 | 0.037 | 2.48 |
| Gemini 2.5 Flash | Enabled | Yes | 38.35% | 0.000192 | 0.384 | 2.65 |
| Gemini 2.5 Pro | Enabled | No | **45.60%** | 0.000080 | 0.160 | 9.34 |
| Gemini 2.5 Pro | Enabled | Yes | 40.33% | 0.001172 | 2.345 | 12.84 |

# 4 Experiments and Results

## 4.1 Experimental Setup

We evaluate on the complete GSM8K test set (1,319 items) and a uniformly sampled subset of PopQA (2,000 items, seed=42). Our baselines are the smallest models in each family without Chain-of-Thought: GPT-4.1-mini and Gemini 2.5 Flash-Lite. We compare two improvement strategies: (i) adding Chain-of-Thought to the baseline model, and (ii) upgrading to a larger model while maintaining reasoning settings.

## 4.2 Complete Performance Results

Tables 1 and 2 present the complete experimental results, showing absolute accuracy, cost, and latency metrics for all model configurations. These raw results reveal striking patterns that become clearer when examined alongside the relative improvements.

## 4.3 GSM8K Results

Table 3 summarizes performance improvements over baseline configurations. For OpenAI models, both Chain-of-Thought and parameter scaling provide substantial accuracy gains, with Chain-of-Thought achieving larger absolute improvements (+49.96 pp vs +11.82 pp) but at higher cost and latency.

For Gemini models, parameter scaling with reasoning disabled achieves substantial gains (+18.52 pp) while actually reducing latency and maintaining very low costs. Most notably, when internal reasoning is enabled (Gemini 2.5 Flash with reasoning), the model achieves 95.36% accuracy without Chain-of-Thought, while adding Chain-of-Thought yields 95.27%—a negligible improvement that comes with substantial cost and latency penalties.

Table 3: GSM8K performance improvements over small No-CoT baseline by family.

| Family | Strategy | ΔAcc (pp) | ΔCost ($) | ΔLatency (s) |
|--------|----------|-----------|-----------|--------------|
| OpenAI | Small + CoT | +49.96 | +0.000721 | +1.64 |
| OpenAI | Upgrade (No-CoT) | +11.82 | +0.000166 | +0.10 |
| Gemini | Small + CoT | +57.18 | +0.000225 | +0.65 |
| Gemini | Upgrade (No reasoning) | +18.52 | +0.000013 | −0.22 |

Table 4: PopQA performance improvements over small No-CoT baseline by family.

| Family | Strategy | ΔAcc (pp) | ΔCost ($) | ΔLatency (s) |
|--------|----------|-----------|-----------|--------------|
| OpenAI | Small + CoT | +4.80 | +0.000179 | +1.35 |
| OpenAI | Upgrade (No-CoT) | +13.50 | +0.000088 | +0.10 |
| Gemini | Small + CoT | +0.09 | +0.000041 | +0.43 |
| Gemini | Upgrade (No reasoning) | +4.00 | +0.000013 | −0.14 |

## 4.4 PopQA Results

Table 4 shows markedly different patterns for knowledge retrieval tasks. Parameter scaling consistently outperforms Chain-of-Thought in both accuracy gains and cost efficiency. More strikingly, for models with reasoning capabilities (GPT-4.1 and Gemini models with reasoning enabled), Chain-of-Thought actually *decreases* accuracy: GPT-4.1 drops from 49.55% to 44.55%, Gemini 2.5 Flash (enabled) drops from 40.13% to 38.35%, and Gemini 2.5 Pro drops from 45.60% to 40.33%.

## 4.5 Cost Efficiency Analysis

Table 5 presents cost-per-percentage-point metrics, revealing clear efficiency patterns. Parameter scaling consistently provides better cost efficiency than Chain-of-Thought, particularly for Gemini models where the difference is substantial (5.6× more efficient for GSM8K).

## 4.6 Pareto Frontier Analysis

Figures 1 and 2 visualize the cost-accuracy Pareto frontiers for both domains. The mathematical reasoning domain shows multiple viable paths to high accuracy, while the knowledge domain demonstrates clear dominance of parameter scaling approaches.

# 5 Discussion

## 5.1 Chain-of-Thought Redundancy in Advanced Models

Our results reveal a critical insight: **Chain-of-Thought becomes largely redundant or even counterproductive when internal reasoning capabilities are available**. This pattern is evident across both domains but manifests differently:

**Mathematical Reasoning with Internal Capabilities.** Gemini 2.5 Flash with reasoning enabled achieves 95.36% accuracy without Chain-of-Thought, while adding Chain-of-Thought yields 95.27%—a negligible difference that comes with substantial cost penalties (23× higher cost per sample). Similarly, Gemini 2.5 Pro shows minimal improvement (96.18% to 96.41%) at dramatically higher cost (25× increase).

**Knowledge Retrieval with Advanced Models.** The pattern is even more pronounced in knowledge tasks, where Chain-of-Thought consistently degrades performance for capable models. This suggests that external reasoning chains can interfere with internal knowledge retrieval processes and introduce paraphrase drift that harms exact-match scoring.

Table 5: Cost efficiency comparison: cost per +1 percentage point (lower is better).

| Domain | Strategy | Cost per +1pp ($) |
|--------|----------|-------------------|
| GSM8K | OpenAI: Small + CoT | $1.44 \times 10^{-5}$ |
| GSM8K | OpenAI: Upgrade | $1.40 \times 10^{-5}$ |
| GSM8K | Gemini: Small + CoT | $3.94 \times 10^{-6}$ |
| GSM8K | Gemini: Upgrade | $\mathbf{7.02 \times 10^{-7}}$ |
| PopQA | OpenAI: Small + CoT | $3.73 \times 10^{-5}$ |
| PopQA | OpenAI: Upgrade | $\mathbf{6.52 \times 10^{-6}}$ |
| PopQA | Gemini: Small + CoT | $4.56 \times 10^{-4}$ |
| PopQA | Gemini: Upgrade | $\mathbf{3.25 \times 10^{-6}}$ |

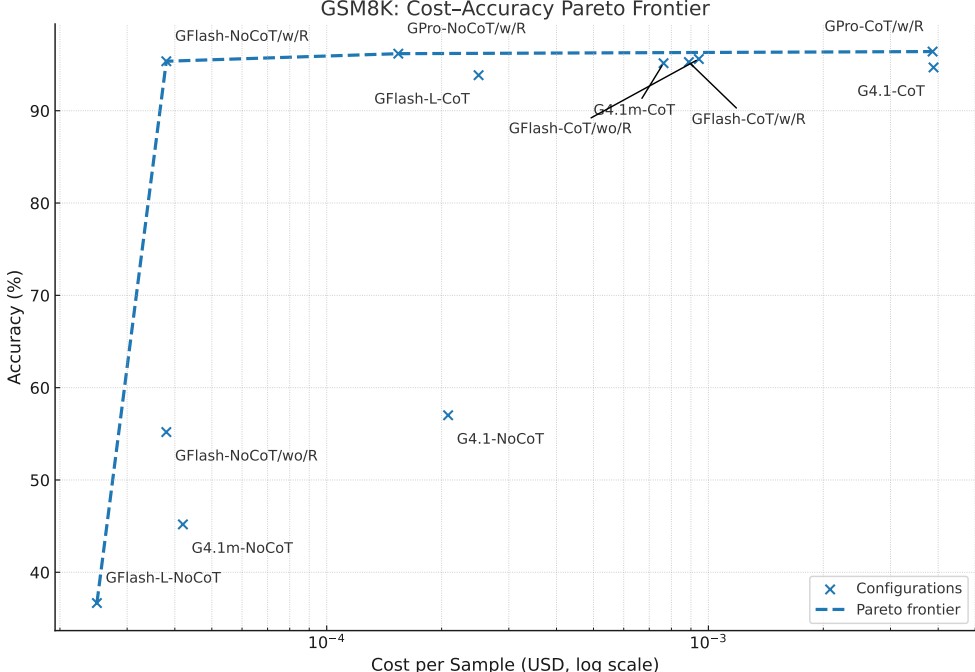

Figure 1: GSM8K cost-accuracy Pareto frontier. Each point represents a (model, CoT, reasoning) configuration plotted by average cost per sample (log scale) versus accuracy on the full test set. The dashed line connects non-dominated configurations. Abbreviations: G4.1m=GPT-4.1-mini, G4.1=GPT-4.1, GFlash-L=Gemini 2.5 Flash-Lite, GFlash=Gemini 2.5 Flash, GPro=Gemini 2.5 Pro, w/R=with internal reasoning, wo/R=without internal reasoning.

## 5.2 Domain-Specific Optimization Strategies

Beyond the redundancy of external reasoning for capable models, our results support distinct optimization strategies:

**Mathematical Reasoning.** Both Chain-of-Thought and internal reasoning address the core problem of computational errors through structured intermediate steps. Chain-of-Thought materializes these steps as tokens (external scratchpad), while internal reasoning processes them in hidden states (latent scratchpad). When tasks admit algorithmic decomposition, both approaches approximate similar computational patterns, explaining their effectiveness on GSM8K.

**Knowledge Retrieval.** Factual QA tasks fail primarily due to missing or imprecise information rather than reasoning errors. Extended reasoning chains can introduce paraphrase drift, where models

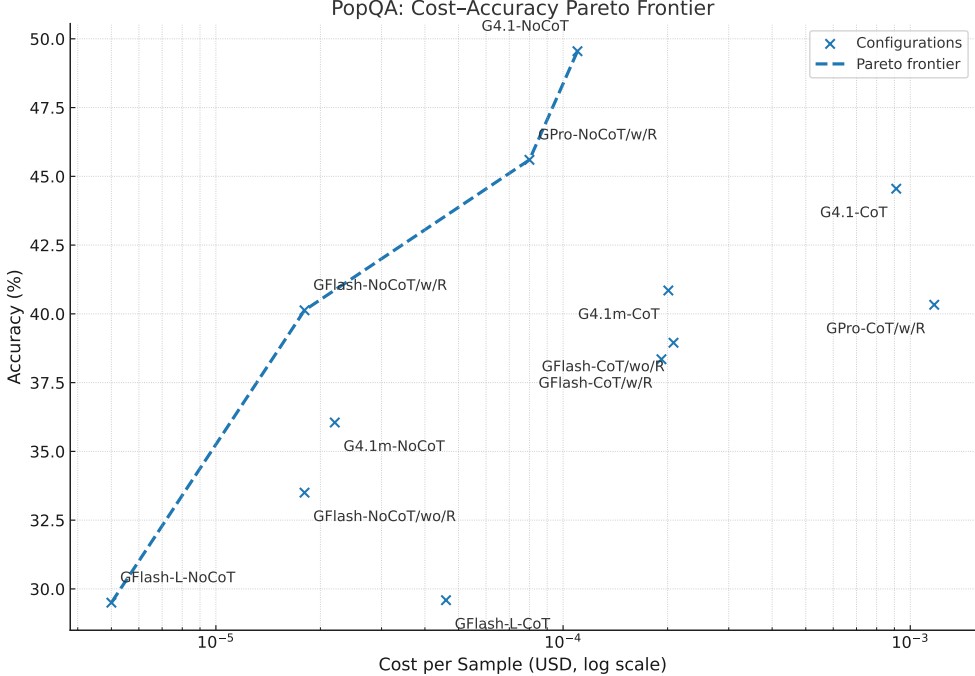

Figure 2: PopQA cost-accuracy Pareto frontier. Parameter scaling configurations dominate the efficient frontier for knowledge retrieval tasks, with Chain-of-Thought approaches generally falling below the optimal cost-accuracy trade-off. Same abbreviations as Figure 1.

subtly alter answer phrasing in ways that harm exact-match scoring. Parameter scaling addresses the root cause by improving factual knowledge, while Chain-of-Thought often adds cost without corresponding benefits.

## 5.3 Practical Deployment Guidelines

Based on our findings, we recommend the following deployment strategy:

1. **Initialize with small models:** Begin with the most cost-effective baseline (small model, no Chain-of-Thought).

2. **Scale parameters first:** Upgrade model capacity before adding reasoning techniques, as this typically provides better cost efficiency and can reduce latency.

3. **Avoid Chain-of-Thought for advanced models:** For models with internal reasoning capabilities, external Chain-of-Thought is typically redundant and can harm performance while increasing costs.

4. **Apply reasoning selectively:** Use Chain-of-Thought primarily for mathematical/logical tasks with smaller models, or when pursuing maximum accuracy regardless of cost.

5. **Consider domain routing:** Implement task-aware routing that applies different strategies based on predicted task characteristics.

## 5.4 Limitations

Several factors limit the generalizability of our findings:

**Dataset scope:** Our PopQA evaluation uses a 2,000-item subset, which may not fully represent the diversity of the complete dataset, particularly for long-tail entities.

**Self-consistency exclusion:** API reliability issues prevented evaluation of self-consistency techniques, which could improve Chain-of-Thought performance, particularly on mathematical tasks.

**Provider-specific features:** Internal reasoning controls are available only for certain models (Gemini in our study), limiting cross-provider comparisons.

**Prompt design:** Our deterministic answer extraction may favor certain approaches; alternative prompt designs could shift absolute performance while preserving relative trends.

# 6  Conclusion

This work provides systematic evidence for optimizing the accuracy-cost-latency trade-offs in agentic LLM systems. Our key insight is that optimal strategies depend critically on both task domain and model capabilities: mathematical reasoning benefits from parameter scaling and reasoning techniques for smaller models, while knowledge retrieval tasks favor parameter scaling alone. Most importantly, for models with internal reasoning capabilities, external Chain-of-Thought becomes largely redundant and can even harm performance.

The practical recommendation is straightforward: begin with cost-effective baselines, prioritize parameter scaling for initial improvements, and avoid external reasoning techniques for advanced models with internal reasoning capabilities. This approach can significantly reduce deployment costs while maintaining or improving accuracy across diverse agentic workflows.

Future work should expand evaluation to additional domains, incorporate self-consistency techniques under stable conditions, and develop automated routing systems that adapt strategies based on real-time task classification and model capability assessment.

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

# A Implementation Details

**Software Environment.** Experiments use Python 3.11 with official API clients for OpenAI and Google. Core dependencies include `pandas`, `numpy`, `tqdm`, and `httpx` for HTTP requests. Complete dependency specifications and environment setup instructions are provided with our code release.

**Reproducibility.** Each experimental run generates detailed JSONL logs containing per-sample inputs, outputs, token counts, latencies, and costs. Aggregated results are exported to CSV format for analysis. All random seeds, API parameters, and reasoning toggles are centrally configured to ensure consistency across conditions.

**Data Availability.** GSM8K uses the standard test split available through HuggingFace datasets. PopQA subset selection uses numpy random sampling with seed=42 for reproducibility. Specific item indices used in our evaluation will be released with our code.

**Code Availability.** Our implementation is available as open source at `https://anonymous.4open.science/r/agent4science-2D92`.

## Agents4Science AI Involvement Checklist

1. **Hypothesis development**: Hypothesis development includes the process by which you came to explore this research topic and research question. This can involve the background research performed by either researchers or by AI. This can also involve whether the idea was proposed by researchers or by AI.

   Answer: [D]

   Explanation: I used Liner's "Hypothesis Generator" agent to propose LLM-related hypotheses that could be executed by AI across the full research pipeline. From several candidates, I selected one and lightly refined it with my own perspective. I then evaluated it with Liner's "Hypothesis Evaluator" agent and incorporated its feedback. Through this iteration, with minimal human steering but significant AI ideation and critique, the final hypothesis used in the paper was produced.

2. **Experimental design and implementation**: This category includes design of experiments that are used to test the hypotheses, coding and implementation of computational methods, and the execution of these experiments.

   Answer: [C]

   Explanation: I used cursor with the claude sonnet-4 model to generate the experiment code. The initial plan was to run open-source models on GPU instances, but the AI-generated code yielded implausible results; my manual "vibe coding" attempts did not fix them. I pivoted to LLM API calls to simplify execution. Even then, many errors remained, so I read the code, diagnosed issues, and guided the coding agent on where and how to patch them. AI produced most of the code, while I performed validation, debugging, and design corrections—hence a rating of C.

3. **Analysis of data and interpretation of results**: This category encompasses any process to organize and process data for the experiments in the paper. It also includes interpretations of the results of the study.

   Answer: [B]

   Explanation: Data analysis and interpretation were primarily done by human, with cursor and chat-gpt-5 assisting. Human verified whether results were valid, flagged anomalies, and decided when to proceed or halt analyses. chat-gpt-5 helped summarize findings, suggest checks, and organize tables/figures, but the key judgments—accepting results, revising analyses, or updating conclusions—were mine. This is why I rate this category as B.

4. **Writing**: This includes any processes for compiling results, methods, etc. into the final paper form. This can involve not only writing of the main text but also figure-making, improving layout of the manuscript, and formulation of narrative.

   Answer: [D]

   Explanation: For the initial draft, I used chat-gpt-5 and claude sonnet-4, supplying the hypothesis, experimental plans, results, and cursor prompts. To refine the draft, I used Liner's Peer Review Agent, incorporated its feedback with claude sonnet-4, and repeated this feedback loop about three times while checking for content drift. I stopped around the fourth iteration when forced changes began to induce hallucinations. Figures were generated by chat-gpt-5 from the results, and citations were suggested by Liner's "Citation Recommender" and integrated into LaTeX. Overall, writing relied heavily on AI agents, with human oversight for correctness and coherence.

5. **Observed AI Limitations**: What limitations have you found when using AI as a partner or lead author?

   Description: The hardest stage was coding and running experiments in cursor. The AI often showed overconfidence—treating incomplete runs as "finished," missing global context, or producing plausible but incorrect outputs. When the code failed semantically (no crash, wrong results), the agent struggled to localize faults. I had to perform root-cause analysis and propose concrete fixes, then direct the agent to implement them. Using AI-generated (rather than human-written) code increased verification overhead. In short: limited end-to-end verification and insufficient epistemic humility were the main pain points.

