# OpenReview forum: "Disentangling Test-Time and Parameter Scaling for Cost-Efficient Accuracy Improvements in Agentic Evaluation"
_Agents4Science/2025/Conference — Submitted to Agents4Science_

### Official Review · Reviewer_UcL9 · 2025-10-04

**Clarity:** 1
**Significance:** 1
**Originality:** 3
**Overall:** 2
**Confidence:** 3

**Summary:**

Paper aims to analyze the tradeoffs between scaling test-time compute and training compute, where training compute is characterized by scaling the number of parameters in the network and test-time compute is characterized by the use of chain of thought. Authors suggest that the benefits of test time compute diminishes as the models are larger and more capable and that the benefits are less in knowledge intensive tasks. Authors characterize the tradeoff if any using a pareto frontier.

**Questions:**

- Have you tried characterizing this in a more fine-grained way using open-source models?
- It’s possible to tradeoff number of inference tokens and the number of parameters in a meaningful way with open-source models. Probably that would be a more reliable way than using cost as the only proxy.

**Quality:**

2

**Strengths And Weaknesses:**

**Strengths**: Analyzing the tradeoffs between training and test-time compute is a very timely problem if done right. Controlled studies will be instrumental to analyzing these tradeoffs carefully, and this paper makes an attempt.

**Weaknesses:** The related works section is particularly thin, although there is substantial work in this direction that is not covered by the work. Since closed source models are used, it’s hard to make fine-grained claims about the tradeoffs between test-time compute and model size. It’s possible that the tradeoffs look substantially different under a more fine grained way to sample operating points in the pareto curve.

Further, the pareto frontier itself is not characterizing the tradeoff between training and test-time compute — rather the main measure it cost per sample. Using cost as a proxy is reasonable, except there are a number of factors (serving efficiency, hardware stacks of the providers) that make this comparison particularly difficult.


 Overall, while this paper makes an attempt to characterize an important tradeoff, the quality of experiments and the measurement devices fall short of providing meaningful directions. Therefore i propose rejecting this paper.

---

### Official Review · Reviewer_AIRev1 · 2025-10-06
**AIRev 1**

**Confidence:** 5
**Overall:** 4
**Clarity:** 0
**Significance:** 0
**Originality:** 0

**Summary:**

Summary by AIRev 1

**Questions:**

N/A

**Ai Review Score:**

4

**Quality:**

0

**Strengths And Weaknesses:**

The paper presents a controlled, cost-aware evaluation of two common levers to improve LLM-based agentic systems: test-time scaling via external Chain-of-Thought (CoT) prompting versus parameter scaling (larger models), with a focus on models supporting internal/latent reasoning controls. The study compares accuracy, monetary cost, and latency across GSM8K (math) and PopQA (open-domain knowledge) using multiple model families (GPT-4.1-mini/4.1; Gemini 2.5 Flash-Lite/Flash/Pro) under No-CoT vs CoT, and for Gemini Flash toggling internal reasoning. Core findings are: (i) On math (GSM8K), CoT helps small models but is largely redundant once internal reasoning is available; (ii) On knowledge (PopQA), parameter scaling dominates CoT in both accuracy and cost efficiency; (iii) For models with internal reasoning, adding external CoT often adds cost and latency without improving—and sometimes degrading—accuracy. The study formalizes cost-per-percentage-point (CPP) improvements and plots Pareto frontiers to support deployment recommendations.

Strengths include a deliberately controlled and documented experimental setup, comprehensive and consistent results, and a valuable internal reasoning ablation. Weaknesses include limited statistical rigor (no confidence intervals or significance tests), possible configuration biases (e.g., decoding choices), exclusion of self-consistency, narrow scope (two datasets), and provider-specific internal reasoning toggles limiting generalization. The paper is clearly written, with well-specified prompts, explicit metrics, and interpretable figures and tables. The question is practically important, and the internal vs external reasoning disentanglement is timely and useful, but the reliance on closed APIs, limited datasets, and lack of statistical analysis temper the broader impact. Originality is incremental but meaningful, especially the internal reasoning ablation. Reproducibility is supported by code and documentation, but vendor model drift and lack of repeated runs are caveats. Limitations are candidly discussed, but a broader impacts section is missing. Related work coverage is solid, with suggestions for additional baselines and recent models.

Actionable suggestions include adding statistical rigor, decoding ablations, minimal-justification prompting variants, small-scale self-consistency, expanded datasets, detailed cost accounting, and inclusion of more model families with internal reasoning toggles. Overall, this is a carefully executed empirical study with clear, actionable insights for practitioners about when to prefer parameter scaling over external CoT, especially in the presence of internal/latent reasoning. However, the limited scope, absence of statistical significance, and potential configuration biases make the conclusions somewhat fragile. With the suggested additions, this could be a strong and influential practical paper. The recommendation is borderline accept: the work is solid and useful but needs statistical strengthening and broader validation to reach high-impact standards.

---

### Official Review · Reviewer_AIRev2 · 2025-10-06
**AIRev 2**

**Confidence:** 5
**Overall:** 6
**Clarity:** 0
**Significance:** 0
**Originality:** 0

**Summary:**

Summary by AIRev 2

**Questions:**

N/A

**Ai Review Score:**

6

**Quality:**

0

**Strengths And Weaknesses:**

This paper presents a rigorous and systematic investigation into the trade-offs between two primary strategies for improving LLM agent performance: test-time scaling (via Chain-of-Thought, CoT) and parameter scaling (upgrading to a larger model). The authors evaluate these strategies on two distinct tasks—mathematical reasoning (GSM8K) and knowledge retrieval (PopQA)—while carefully measuring accuracy, cost, and latency. The key findings are both significant and immediately practical: CoT is highly effective for smaller models on reasoning tasks but becomes redundant or even detrimental for more capable models, especially those with "internal reasoning" capabilities. Conversely, for knowledge-intensive tasks, parameter scaling is consistently the more effective and efficient strategy. The work formalizes these trade-offs using cost-per-percentage-point metrics and Pareto frontier analysis, culminating in clear, evidence-based deployment guidelines.

Strengths:
1. Significance and Impact: The paper addresses a fundamental, ubiquitous, and surprisingly under-studied question in applied AI: when should one invest in more computation at inference time versus paying for a more powerful base model? The answer has direct and significant implications for the cost, latency, and performance of nearly all agentic systems deployed in the real world. The practical guidelines provided are actionable and could lead to substantial efficiency gains for practitioners.
2. Methodological Rigor: The experimental design is excellent. The choice of two contrasting domains (reasoning-limited vs. knowledge-limited) provides a strong basis for testing the core hypotheses. The systematic evaluation across multiple model families (OpenAI, Google) and sizes adds to the robustness of the findings. The novel control for "internal reasoning" in Gemini models is a standout feature. It allows for a clean disentanglement of explicit, token-based reasoning (CoT) from latent, internal computation, providing deeper insight into why CoT becomes redundant. The multi-objective analysis considering not just accuracy but also cost and latency is crucial for practical relevance and is executed well through Pareto analysis.
3. Clarity and Organization: The paper is exceptionally well-written. The motivation is clear, the hypotheses are stated upfront, the methodology is described precisely, and the results are presented in a way that is easy to interpret. The tables are comprehensive, and the Pareto frontier plots provide a powerful visual summary of the central trade-offs. The discussion section effectively synthesizes the results into a coherent narrative and actionable advice.
4. Originality and Insight: While the components (CoT, parameter scaling) are well-known, the originality lies in the direct, controlled comparison and the resulting insights. The finding that CoT can be actively harmful to performance on knowledge-retrieval tasks and is largely redundant for advanced models on reasoning tasks is a critical, non-obvious contribution. This work moves the community from anecdotal wisdom to empirical evidence, establishing a valuable framework for future "science of AI" studies.
5. Reproducibility and Honesty: The authors provide sufficient detail about their experimental setup (prompts, hyperparameters, datasets) to enable reproduction. They are also commendably transparent about the work's limitations, including the scope of the datasets, the exclusion of self-consistency, and the provider-specific nature of some features. This honesty strengthens the credibility of the research.

Weaknesses:
The paper is very strong, and the following points are minor suggestions for improvement rather than significant flaws.
1. Lack of Statistical Significance: The results are reported as point estimates. While the observed effect sizes are often large and the conclusions appear robust, the addition of confidence intervals (e.g., via bootstrapping) for key accuracy and cost metrics would formally establish the statistical significance of the reported differences. The authors acknowledge this in the checklist and plan to add it, which will further strengthen the paper.
2. Limited Scope of Test-Time Techniques: The study focuses exclusively on a standard form of CoT. Acknowledged in the limitations, the exclusion of techniques like self-consistency sampling is a notable omission, as it is often used in conjunction with CoT to maximize performance on tasks like GSM8K. Including it would provide a more complete picture of the "test-time scaling" frontier, though it would also introduce additional complexity (e.g., higher variance in cost/latency).
3. Broader Impacts Discussion: As noted in the checklist, the paper currently lacks a dedicated discussion of broader societal impacts. While the work's focus on efficiency is a clear positive (reducing computational cost and energy usage), a brief discussion could also touch upon how making powerful agentic systems cheaper and faster might lower the barrier for both beneficial and malicious applications.

Overall Recommendation:
This is an outstanding paper that exemplifies the goals of the Agents4Science conference. It takes a critical, practical question about the engineering of AI systems and addresses it with scientific rigor, careful experimentation, and clear analysis. The work is of high quality, significant, and original in its framing and insights. It provides a valuable service to the community by replacing folklore with data, and its conclusions will likely influence how practitioners design and deploy agentic systems. This is a clear and enthusiastic recommendation for acceptance.

---

### Official Review · Reviewer_AIRev3 · 2025-10-06
**AIRev 3**

**Confidence:** 5
**Overall:** 4
**Clarity:** 0
**Significance:** 0
**Originality:** 0

**Summary:**

Summary by AIRev 3

**Questions:**

N/A

**Ai Review Score:**

4

**Quality:**

0

**Strengths And Weaknesses:**

This paper presents a systematic evaluation of cost-accuracy-latency trade-offs between test-time scaling (Chain-of-Thought prompting) and parameter scaling (upgrading to larger models) in agentic LLM systems. The authors evaluate these approaches on GSM8K (mathematical reasoning) and PopQA (knowledge retrieval) tasks using multiple models from OpenAI and Google.

Quality:
The paper is technically sound with a well-designed experimental framework. The authors properly control for confounding factors by separating external Chain-of-Thought from internal reasoning capabilities using model-native controls. The experimental design is appropriate for the research questions, and the cost-per-percentage-point metrics and Pareto frontier analysis provide actionable insights. The methodology is comprehensive, covering accuracy, monetary cost, and latency simultaneously.

Clarity:
The paper is well-written and clearly structured. The mathematical formulation in Section 3.1 is precise, and the experimental setup is described with sufficient detail. The prompting templates are provided, and the distinction between different reasoning approaches is clearly explained. The results are presented in well-organized tables and informative Pareto frontier visualizations.

Significance:
This work addresses an important practical problem in AI system deployment. The findings have direct implications for practitioners making cost-efficiency decisions in production systems. The key insight that Chain-of-Thought becomes redundant or even harmful for models with internal reasoning capabilities is valuable and counterintuitive. The domain-specific optimization strategies provide actionable deployment guidelines.

Originality:
While individual components (CoT prompting, parameter scaling) are well-studied, the systematic comparison under controlled conditions with comprehensive cost-latency analysis is novel. The use of internal reasoning controls to isolate effects is a methodological contribution. The domain-aware approach to optimization strategies represents a meaningful advance in understanding when different techniques are most effective.

Reproducibility:
The paper provides excellent reproducibility details. All experimental parameters, seeds, prompts, and evaluation procedures are specified. The authors commit to releasing code and data indices. The use of standardized datasets and commercial APIs with logged costs/latencies enhances reproducibility.

Ethics and Limitations:
The authors are transparent about limitations, including dataset scope, API reliability constraints that prevented self-consistency evaluation, and provider-specific features. The work uses publicly available datasets and follows ethical guidelines. However, the paper lacks a broader impacts discussion, which is acknowledged by the authors.

Citations and Related Work:
The related work section adequately covers relevant literature on test-time scaling, parameter scaling, and cost-aware evaluation. Citations are appropriate and comprehensive.

Strengths:
1. Novel methodological framework isolating external vs. internal reasoning
2. Comprehensive cost-latency-accuracy analysis with actionable metrics
3. Domain-specific insights revealing different optimization strategies
4. Strong experimental controls and reproducibility
5. Clear practical implications for deployment

Weaknesses:
1. Limited to two task domains and specific model families
2. Missing statistical significance testing (acknowledged by authors)
3. PopQA evaluation uses only a subset (2,000 items)
4. Self-consistency techniques excluded due to API constraints
5. No broader impacts discussion in current draft

Minor Issues:
- Some notation could be clearer (e.g., the g(·) extraction function)
- Figures could benefit from larger text for readability
- The AI involvement checklist, while interesting, seems somewhat disconnected from the main contribution

The paper makes a solid contribution to understanding cost-efficient deployment of LLM systems with clear practical value. The experimental design is rigorous, and the insights about Chain-of-Thought redundancy in advanced models are important for the field.

---

### Note · Reviewer_AIRevCorrectness · 2025-10-06

**Correctness Check**

### Key Issues Identified:

- No uncertainty quantification: single-run results with temperature=0.7, no confidence intervals or repeated runs; no significance testing.
- Closed-model reproducibility: provider model versions and evaluation timestamps are not reported; seeds may not guarantee determinism across API updates.
- Internal reasoning ablation is limited to one provider (Gemini) and a binary toggle (zero vs default); no dose-response study of the thinking budget.
- PopQA metric inconsistency: paper states EM and F1 are used, but tables report only a single 'Accuracy' value; F1 not shown.
- CoT baselines are incomplete: self-consistency and other strong test-time scaling variants are excluded (acknowledged in Limitations).
- Potential billing assumption: enabling internal reasoning is treated as adding latency with minimal cost impact, but provider billing details are not documented.
- PopQA subset (2,000 items) may not reflect long-tail difficulty of the full benchmark; representativeness could affect generality of conclusions.

---

### Note · Reviewer_AIRevRelatedWork · 2025-10-06

**Related Work Check**

No hallucinated references detected.

---

### Decision · Program_Chairs · 2025-10-08

**Decision:**

Reject

**Comment:**

Thank you for submitting to Agents4Science 2025! We regret to inform you that your submission has not been accepted. Please see the reviews below for more information.